# How Cells Die in Psoriasis?

**DOI:** 10.3390/ijms26083747

**Published:** 2025-04-16

**Authors:** Chung-Han Chen, Nan-Lin Wu, Tsen-Fang Tsai

**Affiliations:** 1Department of Education, National Taiwan University Hospital, Taipei City 100, Taiwan; b05401106@ntu.edu.tw; 2Department of Dermatology, MacKay Memorial Hospital, Taipei City 104, Taiwan; alvin.4200@mmh.org.tw; 3Department of Medicine, MacKay Medical College, New Taipei City 252, Taiwan; 4Institute of Biomedical Sciences, MacKay Medical College, New Taipei City 252, Taiwan; 5Department of Dermatology, National Taiwan University Hospital, Taipei City 100, Taiwan

**Keywords:** psoriasis, cell death, prospective therapeutic approaches

## Abstract

Psoriasis, a chronic immune-mediated inflammatory skin disorder characterized by keratinocyte hyperproliferation and inflammatory cell infiltration, involves multiple distinct programmed cell death pathways in its pathogenesis. Following the Nomenclature Committee on Cell Death recommendations, we analyzed the current literature examining diverse modes of cellular death in psoriatic lesions, with particular focus on keratinocyte cell death patterns and their molecular signatures. Analysis revealed several distinct cell death mechanisms: autophagy dysfunction through IL-17A pathways, decreased apoptotic activity in lesional skin, medication targeting anoikis in psoriasis, upregulated necroptosis mediated by RIPK1/MLKL signaling, gasdermin-mediated pyroptosis with enhanced IL-1β secretion, coordinated PANoptotic activation through specialized complexes, PARP1-mediated parthanatos promoting cutaneous inflammation, iron-dependent ferroptosis correlating with Th22/Th17 responses, copper-dependent cuproptosis with elevated MTF1/ATP7B/SLC31A1 expression, and NETosis amplifying immune responses through interaction with the Th17 axis. The intricate interplay between these cell death mechanisms has led to the development of targeted therapeutic strategies, including mTOR inhibitors for autophagy modulation, RIPK1 inhibitors for necroptosis, and various approaches targeting ferroptosis and NETosis, providing new directions for more effective psoriasis treatments.

## 1. Introduction

Psoriasis is a chronic immune-mediated skin disorder characterized by keratinocyte hyperproliferation and inflammatory cell infiltration, manifesting clinically as erythematous plaques with distinctive silvery scales [1]. The hallmark histopathological features include acanthosis, parakeratosis, attenuation of the granular layer, dilated dermal capillaries, and neutrophilic infiltration, reflecting the underlying dysregulation of keratinocyte death processes and immune responses. Historically, psoriasis has been considered as a hyperproliferative disorder of keratinocytes, and studies demonstrate that psoriatic keratinocytes exhibit enhanced resistance to apoptotic induction compared to normal skin cells [2]. Quantitative analyses reveal differential apoptotic indices—0.12% in normal epidermis, 0.035% in established psoriatic lesions, and 0.31% in regressing psoriatic lesions [3]. Thus, the understanding of how cells, particularly keratinocytes, die in psoriasis with epidermal remodeling is important in the natural course and treatment of psoriasis.

Recent advances in cell death research have revealed multiple distinct pathways through which cells undergo programmed or regulated death, including autophagy, apoptosis, anoikis, necroptosis, pyroptosis, PANoptosis, parthanatos, ferroptosis, cuproptosis and NETosis. Each of these programmed cell death pathways exhibits unique molecular signatures and contributes distinctively to the inflammatory cascade associated with psoriasis [4,5,6,7,8,9,10] (Figure 1). Understanding the intricate interplay between these cell death mechanisms and their contributions to psoriatic pathology is fundamental for developing targeted therapeutic interventions. This comprehensive review investigates the diverse modes of cellular death in psoriatic lesions, particularly focusing on keratinocytes, and their implications for disease progression and possible treatment strategies. Although psoriasis is characterized by aberrant cornification, which process was recognized as a mode of cell death by the previous classification of Nomenclature Committee on Cell Death (NCCD) [11], herein we analyze keratinocyte cell death patterns in psoriasis, excluding cornification, which is now classified as terminal differentiation rather than cell death following the latest NCCD recommendations [12].

## 2. Methods

### 2.1. The Approach: Bibliographic Research

We performed a literature search in electronic databases (PubMed, Embase, Google Scholar) for relevant articles from inception to 31 January 2025 utilizing the following keywords: (cell death or autophagy or autophagic cell death or apoptosis or anoikis or necroptosis or pyroptosis or PANoptosis or parthanatos or ferroptosis or cuproptosis or NETosis) AND (psoriasis or interleukin 17 or interleukin 23 or tumor necrosis factor or methotrexate or retinoid or mTOR inhibitor or calcitriol or calcipotriol or ultraviolet or dimethyl fumarate or nicotinamide or glutathione or selenium or coenzyme Q10 or deferoxamine or myeloperoxidase inhibitor or protein arginine deiminase 4 inhibitor). Titles and abstracts were inspected to include articles concerning psoriasis and its treatment or pathogenesis for review. Articles of both human and animal models were included. Only articles available in English were included.

### 2.2. Tools for Manuscript Preparation and Visualization

The Claude artificial intelligence platform was employed to assist in enhancing the quality and fluency of academic text, and all figures were created with BioRender.com.

## 3. Results

### 3.1. Autophagic Cell DEATH

#### 3.1.1. Autophagic Cell DEATH and Psoriasis

Autophagy, a cellular self-degradation mechanism, is an essential biological process that maintains cellular homeostasis through the lysosome-mediated degradation of various cellular components, including nucleic acids, proteins, lipids, and organelles [13]. This fundamental process plays vital roles in cellular differentiation, development, and survival. In the context of psoriasis pathogenesis, autophagic cell death is particularly significant, modulating inflammatory cascades crucial to disease development [14]. The intricate interplay between autophagic cell death and inflammatory pathways is evidenced by multiple studies; Toll-like receptor (TLR)2/6 or TLR4 stimulation activates the autophagic cell death in human keratinocytes [15], while psoriasis-associated cytokines such as TNF-α and IL-17A impair autophagy in these cells [16,17]. Several key inflammatory mediators modulate autophagic function in psoriatic conditions. IL-17A induces autophagic dysfunction in keratinocytes through the PI3K/AKT/mTOR pathway, resulting in elevated pro-inflammatory cytokine production, including IL-6 and IL-8 [18]. Similarly, IL-33 inhibits autophagy via STAT3 phosphorylation, thereby exacerbating the inflammatory response in a psoriatic mouse model [19]. Aryl hydrocarbon receptor (AhR) activation suppresses autophagy and promotes skin inflammation through the NF-Kappa B (NF-κB)/p38 mitogen-activated protein kinase (MAPK) signaling pathway [20], leading to inflammatory cytokine secretion (IL-1β, IL-6, and TNF-α) in keratinocytes [18]. Furthermore, inactivation of the MAPK family decreases keratinocyte autophagy, which correlates positively with psoriatic severity in both patient and mouse models, underscoring the critical role of autophagy in keratinocyte proliferation and differentiation through mechanisms including cell cycle regulation and mitochondrial reactive oxygen species production [21]. These findings suggest that autophagy dysfunction is integrally associated with inflammatory processes and the pathogenesis of psoriasis.

#### 3.1.2. Therapeutic Modulation of Autophagy in Psoriasis

Current therapeutic approaches targeting autophagy pathways show promising results in psoriasis treatment. Retinoids, widely used in patients with psoriasis, normalize keratinization [22] and promote autophagosome maturation and autophagy induction [23], suggesting a potential mechanism through which keratinization may be regulated.

Ultraviolet B (UVB) therapy represents another treatment option for patients with psoriasis, demonstrating the dual benefits of both pruritus relief and immunomodulatory effects [24]. Moreover, studies suggest that UVB stimulates epidermal cell autophagy through the GSK3β/MAPK signaling pathway [25]. Importantly, the therapeutic targeting of these pathways shows promise, as evidenced by rapamycin, an mTOR inhibitor widely used in clinical practice, which has demonstrated therapeutic potential in both cellular and animal models of psoriasis through its inhibition of the mTOR pathway [26,27]. In addition, a case report also showed that the combination of mTOR inhibitors including everolimus and tacrolimus could treat recalcitrant psoriasis [28]. Metformin, a well-established therapeutic agent for type II diabetes mellitus, has demonstrated its efficacy in the treatment of psoriasis [29]. Metformin is also an mTOR inhibitor and promotes the conversion of T helper 17 (Th17) cells to regulatory T (Treg) cells through enhanced autophagic processes [30], suggesting its potential therapeutic application in psoriasis management through the modulation of type 17-mediated inflammation. Consequently, therapeutic interventions targeting IL-17-mediated signaling cascades may represent a promising strategic approach for psoriasis treatment, operating through the restoration of physiological autophagic function and related mechanisms [31].

### 3.2. Apoptosis

#### 3.2.1. Apoptosis and Psoriasis

Apoptosis, the prototype of programmed cell death (RCD), is orchestrated through the activation of caspase family proteases [32]. This process manifests through two principal pathways: intrinsic and extrinsic apoptosis. The intrinsic pathway can be triggered by diverse stimuli, including DNA damage, ribotoxic stress response, endoplasmic reticulum stress, oxidative stress, growth factor deprivation, and microtubular alterations [32,33]. A pivotal step in intrinsic apoptosis involves the activation of pro-apoptotic BCL2 family effectors—specifically BAX, BAK, and potentially BOK—which facilitate apoptosome formation and the sequential activation of caspase 9 and executioner caspases 3 and 7 [34]. The extrinsic pathway initiates through death receptor–ligand interactions (such as FASL-FAS, TNF-TNFR), forming the death-inducing signaling complex (DISC), which activates caspase 8. Subsequently, caspase 8 triggers cell death either directly via caspase 3/7 or through BH3-interacting domain death agonist-mediated mitochondrial outer membrane permeabilization [34].

Psoriatic skin lesions exhibit complex apoptotic regulation involving multiple protein families and signaling cascades. The Bcl-2 family proteins, comprising both pro-apoptotic (Bax, Bak, Bad) and anti-apoptotic (Bcl-2, Bcl-xL) members, are integral to this regulation. A previous study has demonstrated elevated Bcl-xL expression in psoriatic epidermis compared to normal tissue [35]. TNF-α signaling demonstrates a complex dual role by upregulating both pro- and anti-apoptotic factors [36], ultimately conferring the hyperproliferation of keratinocytes [37]. Furthermore, psoriatic lesions exhibit decreased apoptotic activity compared to normal epidermis [3], with inflammatory mediators IL-15 [38] and TGF-α [39] serving as potent inhibitors of keratinocyte apoptosis observed in psoriatic tissue.

#### 3.2.2. Therapeutic Modulation of Apoptosis in Psoriasis

Current therapeutic interventions demonstrate differential effects on apoptotic pathways. Notably, enhanced apoptotic activity has been observed following the administration of psoralen plus ultraviolet A (PUVA) therapy to psoriatic lesions [3]. Narrowband ultraviolet B (nbUVB) phototherapy has emerged as a widely adopted treatment modality for inflammatory skin conditions. Previous studies have demonstrated that nbUVB induces apoptosis not only in keratinocytes [40,41], but also in T cells [42].

Methotrexate (MTX) is one of the most widely used systemic medications for the treatment of psoriasis. MTX-induced epidermal necrosis is well-documented in psoriasis [43], but rarely reported in other inflammatory diseases such as atopic dermatitis [44]. This disparity largely stems from methotrexate’s mechanism of inducing apoptosis in proliferating keratinocytes [45], coupled with the significantly higher keratinocyte turnover rate in psoriasis compared to atopic dermatitis [46]. Methotrexate intoxication demonstrated a significant elevation in the gene expression of apoptotic biomarkers, namely, Bax, and a significant reduction in antiapoptotic marker Bcl2, as compared to the control value [47]. Furthermore, MTX induces apoptosis through oxidative stress pathways by reducing nitric oxide and increasing caspase-3 levels, partially explaining its therapeutic efficacy in treating psoriatic acanthosis [48]. Contemporary molecular research has unveiled microRNA regulation as a pivotal mediator of MTX’s therapeutic action. MTX administration markedly suppresses miR-155 expression in psoriatic lesions, with mechanistic studies in HaCaT keratinocytes establishing miR-155’s role as a critical determinant of cellular fate. The overexpression of miR-155 significantly modulates cell cycle progression, manifesting in G0/G1 phase arrest and the inhibition of apoptotic processes [49].

Vitamin D demonstrates a concentration-dependent biphasic effect on keratinocyte survival: physiological levels elicit protection against multiple apoptotic triggers, including ceramide, UV radiation, and TNF-α, whereas elevated concentrations promote programmed cell death [50]. Furthermore, vitamin D exerts its antiapoptotic effects through multiple molecular cascades, encompassing the modulation of the pro-survival to pro-death protein balance, specifically enhancing the ratio of Bcl-2 to Bad and Bax [51]. Calcipotriol, a vitamin D3 analogue widely used in topical psoriasis treatment, has been demonstrated to enhance apoptosis in human psoriatic keratinocytes [52]. Retinoids have been extensively validated as an effective systemic therapeutic approach for psoriasis [53]. They serve crucial roles in vital biological processes, including fetal morphogenesis, cellular differentiation, and apoptosis [54]. In vitro studies utilizing cultured keratinocytes have demonstrated that retinoids induce apoptotic processes [55]. The therapeutic efficacy of retinoids in psoriasis treatment may be attributed to their ability to induce keratinocyte apoptosis.

### 3.3. Anoikis

#### 3.3.1. Anoikis and Psoriasis

Anoikis is a form of apoptotis triggered by detachment from the extracellular matrix (ECM) [56,57]. As a caspase-dependent process, anoikis activates caspase family proteinases, initiating a cascade that leads to cell death similar to apoptotic pathways. While sharing features with apoptosis, anoikis demonstrates unique signaling mechanisms following ECM detachment, involving integrin-mediated signaling, PI3K-AkT signaling cascade, and Fas-dependent pathways [8]. In psoriasis, a hyperproliferative dermatosis, the normal polarized distribution of integrin expression on keratinocytes becomes disrupted [58].

#### 3.3.2. Therapeutic Modulation of Anoikis in Psoriasis

The integrin-dependent mechanism of anoikis offers a potential therapeutic target for intervention. Integrins mediate ECM signaling and serve as regulatory points in the anoikis pathway. This therapeutic approach has led to the development of integrin-targeting agents, including efalizumab and etaracizumab, evaluated in clinical trials for psoriasis treatment [59]. However, efalizumab was subsequently withdrawn from the pharmaceutical market in 2009, following its initial regulatory approval, due to an elevated risk of progressive multifocal leukoencephalopathy [60]. Similarly, the clinical development of etaracizumab was discontinued following the completion of a Phase II randomized, double-blind, placebo-controlled clinical trial investigating its efficacy in psoriasis treatment.

### 3.4. Necroptosis

#### 3.4.1. Necroptosis and Psoriasis

Necroptosis, a regulated form of necrotic cell death, is orchestrated through the activation of tumor necrosis factor receptor (TNFR) by TNFα [61] or FASL/FAS signaling [62]. These receptors subsequently recruit adapter proteins, including TNFR1-associated death domain protein (TRADD) or Fas associated death domain protein (FADD), which interact with RIPK1 and caspase-8 or -10 [63]. This process involves the molecular interplay of receptor-interacting protein kinase 1 (RIPK1), receptor-interacting protein kinase 3 (RIPK3), and mixed lineage kinase domain-like pseudokinase (MLKL), which has been implicated in various inflammatory conditions. The RIPK1/RIPK3 complex initiates MLKL phosphorylation and activation, leading to oligomerization and necrosome formation [64]. These structures subsequently traffic to the plasma membrane in association with tight junction proteins, where they accumulate to form micron-sized structures. The MLKL oligomers specifically localize to phosphatidylinositol phosphate (PIP)-rich regions in the plasma membrane, where they form large pores. The formation of these MLKL pores ultimately triggers necroptotic cell death through multiple mechanisms: facilitating ion influx, inducing cell swelling, and promoting membrane lysis [65]. This cascade of events culminates in the uncontrolled release of intracellular material.

A recent study has demonstrated necroptosis’s crucial role in psoriasis pathogenesis, evidenced by the significant upregulation of RIPK1 and MLKL throughout all epidermal layers in human psoriatic lesions [66]. This necroptotic activation has been further validated in imiquimod (IMQ)-induced psoriasiform murine models. Pharmacological inhibition using R-7-Cl-O-Necrostatin-1 (Nec-1s) and necrosulfonamide (NSA), targeting RIPK1 and MLKL, respectively, effectively suppressed necroptosis in both HaCaT cells and IMQ mouse models. This intervention simultaneously attenuated IMQ-induced inflammatory responses and significantly reduced the production of key inflammatory mediators, including IL-1β, IL-6, IL-17A, IL-23A, CXCL1, and CCL20 [66]. The suppression of type 17-associated cytokines, particularly IL-17A and IL-23A, through necroptosis inhibition suggests that necroptosis may contribute to psoriasis pathogenesis via type 17 inflammation. These findings present compelling evidence for targeting necroptosis as a potential therapeutic strategy in psoriasis treatment.

#### 3.4.2. Therapeutic Approaches Targeting Necroptosis in Psoriasis

Dimethyl fumarate (DMF), a derivative of fumaric acid esters, has emerged as a significant first-line systemic therapy in some countries for moderate-to-severe plaque psoriasis [67,68]. Clinical observations indicate that DMF typically ameliorates skin inflammation within the initial three months of treatment [69]. However, the precise molecular mechanisms underlying DMF’s therapeutic effects in psoriasis remain incompletely understood. Recent evidence suggests that DMF could exert therapeutic effects through the inhibition of the RIPK1–RIPK3–MLKL necroptotic signaling axis, as demonstrated in both mice models and cellular systems [70]. Additionally, saracatinib, a dual Src/Abl kinase inhibitor currently under clinical development for the treatment of Parkinson’s disease, psychosis, idiopathic pulmonary fibrosis and fibrodysplasia ossificans progressiva, demonstrated therapeutic potential in an IMQ-induced psoriasis model by suppressing MLKL phosphorylation and subsequent necroptotic cell death [71].

### 3.5. Pyroptosis

#### 3.5.1. Pyroptosis and Psoriasis

Pyroptosis, a distinctive form of programmed cell death, is primarily mediated through the gasdermin protein family [72]. The molecular mechanisms of pyroptosis encompass multiple pathways. In the canonical pathway, Pathogen-Associated Molecular Patterns (PAMPs) and Damage-Associated Molecular Patterns (DAMPs) initiate intracellular signaling cascades, leading to the assembly of inflammasomes with pro-caspase-1, subsequently activating caspase-1. Activated caspase-1 cleaves both Gasdermin D (GSDMD) and pro-IL-1β/18. The N-terminal fragment of GSDMD (N-GSDMD) forms nonselective pores in the cell membrane, facilitating water influx that culminates in cell lysis and death while enabling IL-1β and IL-18 secretion. The noncanonical pathway involves the cytosolic lipopolysaccharide (LPS) activation of caspase-4/5, triggering pyroptosis through GSDMD cleavage [73]. GSDMD cleavage induces potassium efflux, which mediates NLRP3 inflammasome assembly and the subsequent processing of pro-IL-1β and pro-IL-18. IL-1β facilitates Th17 cell differentiation and activation [74], while IL-18 stimulates both Th17 and γδ T cells, promoting IL-17 secretion [75].

Recent research has elucidated that GSDMD-mediated pyroptosis exhibits significant pro-inflammatory characteristics, characterized by the substantial release of pro-inflammatory mediators, such as IL-1 [76]. In psoriatic conditions, research has demonstrated the elevated expression of GSDMD in psoriatic skin lesions of humans [77]. Enhanced cleavage products of caspase-1, GSDMD, and IL-1β were observed in both imiquimod-induced psoriasis-like dermatitis (IIPLD) mouse epidermis and M5 (simulating psoriatic inflammatory challenge)-treated keratinocytes in vitro. The critical role of GSDMD in pyroptosis was further validated through genetic studies, where Gsdmd−/− keratinocytes failed to exhibit pyroptotic morphology under M5 stimulation [78]. GSDME can be activated through TNF-α and TNF receptor signaling pathways, which contribute to pyroptotis [79]. Additionally, GSDME has emerged as another key mediator of pyroptosis in psoriatic conditions, with elevated expression observed in keratinocytes. The GSDME-mediated pyroptosis of keratinocytes leads to the secretion of inflammatory mediators, including IL-1β and IL-18 [80]. *Gsdme−/−* mice and caspase-3 inhibitor treatments have demonstrated attenuated skin inflammation and reduced inflammatory cytokine expression [81]. Significantly higher GSDME expression levels have also been documented in human psoriatic lesions compared to healthy controls [82]. Notably, emerging evidence has revealed potential mechanistic interconnections between RSR and pyroptosis [33].

#### 3.5.2. Potential Therapeutic Approaches Targeting Pyroptosis in Psoriasis

Disulfiram is a medication that has historically been used in the treatment of alcohol use disorder, addictions, infections, and inflammatory conditions [83]. Clinical studies have demonstrated the superior efficacy of disulfiram compared to placebo in treating nickel dermatitis [84,85], although the historical literature has presented conflicting evidence regarding the therapeutic potential of disulfiram in psoriasis treatment [86]. While the precise mechanism of disulfiram’s anti-inflammatory action remains incompletely understood, recent investigations have elucidated the efficacy of disulfiram in IMQ-induced psoriasis models [87]. Research demonstrates that while disulfiram permits the processing of IL-1β and Gasdermin D (GSDMD), it significantly inhibits pore formation, thereby preventing IL-1β release and subsequent pyroptotic cell death [88]. In addition, recently developed disulfiram-loaded lactoferrin nanoparticles could be used to alleviate inflammatory diseases such as ulcerative colitis in murine model. Furthermore, recently developed disulfiram-loaded lactoferrin nanoparticles have shown promise in alleviating inflammatory diseases, as demonstrated in a murine model of ulcerative colitis [89]. The newly identified role of disulfiram in GSDMD inhibition, combined with these novel delivery techniques, suggests promising therapeutic applications for this established drug in the management of psoriasis.

### 3.6. PANoptosis

PANoptosis represents a novel inflammatory programmed cell death mechanism, characterized by the coordinated interplay of pyroptosis, apoptosis, and necroptosis pathways through specialized PANoptosome complexes [5]. The molecular architecture of PANoptosomes encompasses three fundamental protein categories: pattern recognition receptors (including ZBP1 and NLRP3) that detect pathogen- or damage-associated molecular patterns, adaptor molecules (such as FADD), and effector proteins (RIPK1, RIPK3, Caspase 1, and Caspase 8) [90]. While PANoptosome composition varies by stimulus, core regulatory proteins essential for executing the three death pathways remain consistent. This distinctive death modality appears in multiple pathological conditions, including Candida albicans and Aspergillus fumigatus infection [91], as well as in various disease states such as cancer [92], organ failure [93], and inflammatory bowel disease [94]. Although correlations between psoriasis and various programmed cell death mechanisms have been studied, the role of PANoptosis in psoriasis remains underexplored.

Transcriptomic analyses of psoriatic lesions have revealed enhanced PANoptotic signaling characterized by the upregulation of key mediators including Caspase-1, NLRP3, GSDMD, and IL-1β compared to non-lesional skin [87]. A comparative transcriptomic study between psoriasis patients and healthy controls has demonstrated the elevated expression of PANoptotic activators including absent in melanoma-2 (AIM2) and interferon regulatory factor 1 (IRF1) [95]

Another analysis of the PANoptosis signatures reveals distinct cellular correlations within the psoriatic microenvironment. Robust positive correlations have been identified with multiple immune and tissue-resident cell populations, including macrophages, dendritic cells, mesenchymal stem cells, Th1 cells, Th2 cells, melanocytes, monocytes, neutrophils, basophils, and keratinocytes. Although conventional research has predominantly focused on lymphocytes, neutrophils, keratinocytes, and dendritic cells as primary mediators in psoriasis pathogenesis, accumulating evidence indicates essential regulatory roles for fibroblasts and mast cells, despite their negative correlation with PANoptotic signatures [96]. The IL-17 signaling pathway shows significant enrichment within the psoriasis PANoptosis signatures [96].

### 3.7. Parthanatos

#### 3.7.1. Parthanatos and Psoriasis

Parthanatos represents a regulated cell death cascade characterized by sequential molecular events, including poly (ADP-ribose) polymerase 1 (PARP-1) hyperactivation, poly (ADP-ribose) (PAR) polymer accumulation, the PAR-mediated recruitment of apoptosis-inducing factor (AIF), mitochondrial AIF release, nuclear translocation of the AIF/macrophage migration inhibitory factor (MIF) complex, and subsequent MIF-dependent large-scale DNA fragmentation [97]. PARP1-mediated parthanatos has been implicated in psoriasis pathogenesis, wherein PARP1 promotes cutaneous inflammation through the induction of parthanatos-mediated cell death [9].

#### 3.7.2. Potential Therapeutic Approaches Targeting Parthanatos in Psoriasis

Nicotinamide (NAM) and its derivative nicotinamide mononucleotide (NMN) function as PARP-1 inhibitors. Previous investigations have demonstrated that NMN exhibits protective effects against IMQ-induced psoriatic inflammation [98]. Furthermore, clinical research has also established the therapeutic efficacy of topical NAM administration in psoriasis treatment [99].

### 3.8. Ferroptosis

#### 3.8.1. Ferroptosis and Psoriasis

Ferroptosis represents an iron-dependent programmed cell death characterized by phospholipid peroxidation. This process integrates multiple cellular metabolic pathways, including redox homeostasis, iron metabolism, mitochondrial function, and the metabolism of amino acids, lipids, and carbohydrates, along with various disease-relevant signaling cascades [100]. Ferroptotic cell death requires three essential components: transition metal iron, reactive oxygen species (ROS), and phospholipids containing polyunsaturated fatty acid chains (PUFA-PLs) [101]. The process is intricately regulated by both cellular metabolism, which influences ROS and PUFA levels, and various extracellular factors. A critical regulator of ferroptosis is selenium, an essential micronutrient required for the biosynthesis of ROS-scavenging selenoproteins, particularly glutathione peroxidase 4 (GPX4) [102]. GPX4 serves as a key inhibitor of phospholipid peroxidation, while cystine uptake through the system Xc⁻ cystine/glutamate antiporter provides additional protection against ferroptosis by supporting GPX activity [103]. The molecular mechanisms underlying ferroptosis involve multiple interconnected pathways. Central to this process is the inhibition of system Xc⁻, a membrane transport complex composed of SLC7A11 and SLC3A2, which normally facilitates the exchange of extracellular cystine for intracellular glutamate [104]. The inhibition of this system impairs cellular cystine uptake, leading to reduced glutathione (GSH) synthesis. Concurrently, cellular iron homeostasis plays a crucial role through the uptake of Fe^3^⁺ and its subsequent reduction to Fe^2^⁺ via the Fenton reaction, generating substantial reactive oxygen species. The ferroptotic cascade is further modulated by GPX4, a GSH-dependent antioxidant enzyme that typically neutralizes toxic lipid peroxides by converting them to their corresponding alcohols [6]. Either the direct inhibition of GPX4 activity or the depletion of its essential cofactor GSH results in the accumulation of lethal lipid peroxides. Recent studies have also illuminated the role of p53 in ferroptosis regulation through its transcriptional suppression of SLC7A11, which compromises cellular cystine uptake.

Notably, similar ferroptotic patterns have been observed in both erastin-treated human primary keratinocytes and the imiquimod (IMQ)-induced psoriasis model. Single-cell analysis has revealed a significant correlation between lipid oxidation activity and the Th22/Th17 response in keratinocytes. Furthermore, ferrostatin-1 (Fer-1), a potent inhibitor of lipid peroxidation, has demonstrated therapeutic potential by suppressing ferroptosis-related changes in erastin-treated keratinocytes and alleviating psoriasiform dermatitis in IMQ-induced models. Significantly, Fer-1 exhibited broad anti-inflammatory effects both in vitro and in vivo, reducing the production of multiple inflammatory cytokines, including TNF-α, IL-6, IL-1α, IL-1β, IL-17, IL-22, and IL-23. This sophisticated interplay between iron metabolism, lipid peroxidation, and antioxidant defense systems underscores the complexity of ferroptotic cell death mechanisms and their potential therapeutic implications in inflammatory skin conditions [105].

#### 3.8.2. Potential Therapeutic Approaches Targeting Ferroptosis in Psoriasis

A published case report demonstrated significant improvement in the Psoriasis Area and Severity Index (PASI) score following intravenous glutathione administration in a patient with scalp psoriasis [106]. Similarly, another case report documented substantial improvement in PASI scores after dietary supplementation with glutathione-enhancing nondenatured whey protein isolate [107]. These clinical observations suggest a potential therapeutic role for glutathione in modulating psoriatic disease activity, warranting further investigation into glutathione-based interventions for psoriasis management. Selenium plays a crucial protective role in modulating oxidative stress, attenuating lesion development, and regulating immune responses in patients with psoriasis [108]. Recent investigations have elucidated that selenium’s anti-ferroptotic properties are intricately linked to the temporal regulation of selenoprotein GPX4 expression, representing a direct and expeditious protective mechanism against lipid peroxidation [109]. The biological activity of selenium is primarily executed through its incorporation into selenoproteins, which serve as critical mediators in ferroptosis inhibition. Coenzyme Q10, an essential antioxidant that reduces oxidative stress, demonstrates therapeutic efficacy in patients with psoriasis [110,111]. Notably, Coenzyme Q10’s interaction with ferroptosis suppressor protein 1 has been shown to suppress ferroptotic cell death, suggesting its potential role in ferroptosis inhibition pathways [112]. Deferoxamine is a chelating agent that binds to iron and facilitates the removal of excess iron from the body. Several studies have demonstrated that iron overload can trigger psoriasisform skin inflammation [113,114]. Furthermore, experimental evidence indicates that deferoxamine reduces keratinocyte proliferation and attenuates epidermal thickness in human 3D organotypic skin models [114]. These findings collectively suggest that deferoxamine may possess therapeutic potential for the treatment of psoriasis.

### 3.9. Cuproptosis

Cuproptosis, identified by Tsvetkov et al. in 2022, represents a copper-dependent programmed cell death pathway characterized by mitochondrial copper ion accumulation [115]. This process triggers the aggregation of lipoylated dihydrolipoamide S-acetyltransferase (DLAT), a crucial component of the mitochondrial tricarboxylic acid (TCA) cycle, thereby precipitating proteotoxic stress that ultimately culminates in cell death [115]. The biological signature of this death pathway is marked by two distinctive features: the specific aggregation of lipoylated mitochondrial enzymes, particularly DLAT, and the concurrent degradation of iron–sulfur cluster (Fe-S)-containing proteins [116].

Notably, transcriptomic analyses have revealed the significant upregulation of cuproptosis-associated genes (MTF1, ATP7B, and SLC31A1) in psoriatic patients [10], while additional clinical research has demonstrated elevated serum copper levels in individuals with psoriasis [117]. Moreover, cellular copper uptake is primarily regulated through the IL-17-STEAP4 axis [118], with elevated STEAP4 expression observed in keratinocytes of patients with generalized pustular psoriasis [119]. Indeed, serum STEAP4 levels are consistently higher in psoriasis patients compared to healthy controls [120], suggesting a potential mechanistic link between cuproptosis and the IL-17 inflammatory cascade. Although the precise mechanisms underlying these associations remain largely unelucidated, and studies investigating copper chelation in psoriasis treatment are currently lacking, the established correlation between cuproptosis, elevated copper concentrations, and psoriasis pathogenesis suggests that therapeutic interventions targeting the cuproptosis pathway may represent a promising novel strategy for psoriasis treatment.

### 3.10. NETosis

#### 3.10.1. NETosis and Psoriasis

Neutrophil extracellular traps (NETs) are released by activated neutrophils in response to various stimuli. While initially discovered in neutrophils, NET formation has subsequently been observed in other innate immune cells, including macrophages, monocytes, mast cells, basophils, dendritic cells, and eosinophils [121]. This process, known as NETosis, is mediated by protein-arginine deiminase 4 (PAD4) and involves the release of intracellular granule components that capture and destroy diverse pathogens, including viral, fungal, bacterial, and protozoal organisms [122]. During NETosis, neutrophils enhance their antimicrobial capabilities through the release of NETs, which comprise extracellular chromatin decorated with histones and various granular proteins [123]. While NETosis primarily functions as a critical host defense mechanism against pathogens, its implications extend beyond microbial control. Recent evidence has elucidated the complex role of NETosis in psoriasis pathogenesis, particularly through its interaction with the Th17 inflammatory axis [124]. In psoriatic conditions, NETosis initiates and amplifies immune responses through multiple pathways. The LL37–DNA complexes generated during NETosis stimulate Th17 cells to secrete cytokines, which subsequently promote keratinocyte LL37 production, establishing a positive feedback loop. Furthermore, these LL37–DNA complexes activate plasmacytoid dendritic cells (pDCs) via TLR9 receptors [125], leading to TNFα production and monocyte activation. The resulting IL-23 secretion further stimulates Th17 cells to produce IL-17A, which reactivates neutrophils and amplifies inflammatory responses. Concurrent IL-8 release recruits additional neutrophils to lesion sites, perpetuating the inflammatory cascade [126]. Clinical observations support the significance of this mechanism, with NETosis detected in nearly all psoriasis skin specimens, predominantly in the epidermis. Notably, the absence of visible NETosis in two cases correlated with milder disease presentation and reduced peripheral blood NETotic cell counts, suggesting a potential correlation between NETosis intensity and disease severity [127].

#### 3.10.2. Potential Therapeutic Approaches Targeting NETosis in Psoriasis

Myeloperoxidase inhibition attenuates psoriasis severity in murine models when administered either systemically or topically [128]. As a result, the myeloperoxidase inhibitor could serve as a potential medication for psoriasis. Previous research has shown that protein arginine deiminase 4 (PAD4) inhibitors can effectively prevent NET formation in arthritis mouse models [129]. Additionally, serum PAD4 levels have been observed to decrease following treatment with anti-IL-17A, anti-TNFα, and methotrexate [130]. These findings suggest that PAD4 inhibitors may represent a promising therapeutic approach for psoriasis treatment.

### 3.11. Summary of Potential Therapeutic Approaches in Psoriasis

The medical management of psoriasis associated with various forms of programmed cell death—including autophagy, apoptosis, anoikis, necroptosis, pyroptosis, parthanatos, ferroptosis, and NETosis—is summarized in Table 1.

## 4. Discussion

This comprehensive review elucidates the intricate interplay between cell death pathways in psoriasis pathogenesis and the therapeutic implications. A pivotal finding demonstrates the critical role of autophagy dysfunction in psoriasis, wherein IL-17A mediates autophagy suppression via the PI3K/AKT/mTOR pathway, establishing a fundamental link between inflammatory signaling and cellular homeostasis. This mechanism not only substantiates the efficacy of mTOR inhibitors, such as rapamycin, but also provides a theoretical framework for developing targeted interventions that modulate autophagic processes. The observation of attenuated apoptotic activity in psoriatic lesions, coupled with upregulated anti-apoptotic factors including Bcl-xL, suggests that impaired programmed cell death contributes substantially to the characteristic hyperproliferative state, providing mechanistic insight into the therapeutic efficacy of apoptosis-enhancing interventions, notably PUVA therapy, methotrexate, vitamin D and retinoids. Furthermore, anoikis, resulting from integrin disruption, has been implicated in psoriasis pathogenesis. Although integrin-targeting therapeutics have faced limited clinical implementation due to leukoencephalopathy complications, this pathway warrants further investigation for therapeutic development. A significant advancement in the field stems from the identification of enhanced necroptosis signaling through RIPK1 and MLKL in affected tissues, where the successful attenuation of inflammation via necroptosis inhibition in experimental models presents promising therapeutic opportunities, particularly through the targeted intervention of the RIPK1–RIPK3–MLKL axis. The involvement of pyroptosis, orchestrated by GSDMD and GSDME, introduces additional complexity to the pathogenic cascade, with elevated expression of these proteins, coupled with their association with IL-1β and IL-18 secretion, illuminating novel aspects of the inflammatory pathway and facilitating innovative therapeutic approaches such as the repurposing of disulfiram as a GSDMD inhibitor. The emergence of PANoptosis as a coordinated mechanism represents a paradigm shift in understanding inflammatory cell death. Strong correlations between PANoptotic signatures and various immune cell populations suggest its central role in orchestrating the inflammatory microenvironment of psoriasis. The contribution of ferroptosis, particularly its association with Th22/Th17 responses, provides novel insights into the influence of iron metabolism and lipid peroxidation, substantiating the therapeutic potential of antioxidants and iron chelators in disease management. Cuproptosis and its connection through elevated copper levels and the IL-17-STEAP4 axis presents new intervention possibilities, while the role of NETosis in amplifying immune responses through interaction with the Th17 axis demonstrates the sophisticated interplay between innate and adaptive immunity, with the correlation between NETosis intensity and disease severity suggesting its potential utility as both a biomarker and a therapeutic target. These insights collectively indicate that effective therapeutic strategies may require the simultaneous modulation of multiple death pathways, as the efficacy of current therapeutic interventions may be attributed to their impacts on various cell death mechanisms rather than the modification of a single pathway.

Future research priorities should encompass the development of combination therapies targeting multiple death pathways, the investigation of the temporal sequence of death mechanisms during disease progression, the identification of pathway-specific biomarkers for personalized treatment approaches, and the exploration of novel therapeutic agents targeting emerging pathways such as cuproptosis, while acknowledging current methodological limitations including the challenge of distinguishing between primary and secondary effects of pathway activation and the potential oversight of significant synergistic effects due to the predominant focus on individual pathways rather than their interactions.

## 5. Conclusions

In conclusion, this extensive analysis elucidates the complex interplay of multiple programmed cell death mechanisms in psoriasis pathogenesis. The identification of these distinct death pathways and their molecular signatures provides crucial insights into the multifaceted nature of psoriatic inflammation. The demonstrated involvement of autophagy dysfunction, apoptotic dysregulation, and various forms of regulated cell death, including necroptosis, pyroptosis, and the newly characterized cuproptosis, underscores the complexity of cellular death processes in psoriatic lesions. Understanding these intricate mechanisms has facilitated the development of targeted therapeutic approaches. These findings not only advance our understanding of psoriasis pathophysiology, but also establish a foundation for novel therapeutic strategies. Future research should focus on elucidating the temporal dynamics and cross-talk between these death pathways, potentially leading to more effective combination therapies for enhanced clinical outcomes in psoriasis management.

## Figures and Tables

**Figure 1 ijms-26-03747-f001:**
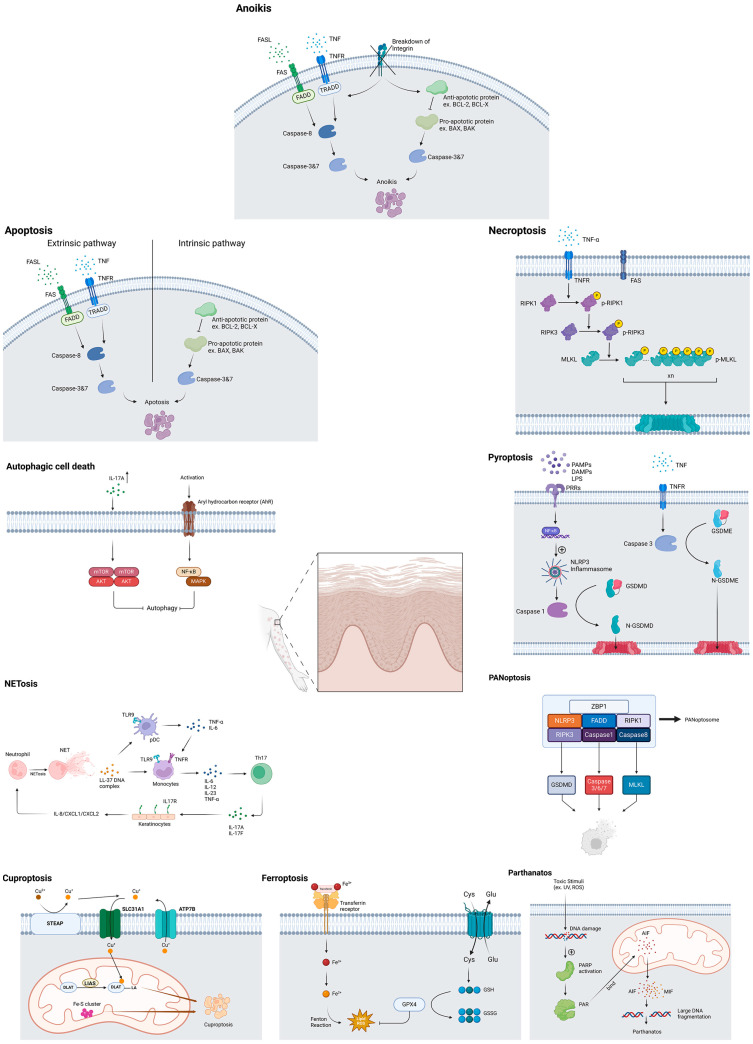
Cellular death mechanisms in psoriasis.

**Table 1 ijms-26-03747-t001:** Cellular death mechanisms potentially implicated in the action of anti-psoriatic therapeutic interventions.

Management	Possible Mechanism	References
Narrowband ultraviolet B (NbUVB) phototherapy	Autophagy, Apoptosis	Y. Yang, et al., 2012 [25] B. M. Aufiero, et al., 2006 [40] S. C. Weatherhead, et al., 2011 [41] M. Ozawa, et al., 1999 [42]
Rapamycin	Autophagy	M. Gao and X. Si, 2018 [26] C. Bürger, et al., 2017 [27]
Everolimus and tacrolimus	Autophagy	K. C. Wei and P. C. Lai, 2015 [28]
Retinoids	Autophagy, Apoptosis	Y. Rajawat, et al., 2010 [23] T. C. Islam, et al., 2000 [55]
Metformin	Autophagy	Z. Huang, et al., 2023 [29]
Psoralen plus ultraviolet A (PUVA)	Apoptosis	M. Laporte, et al., 2000 [3]
Methotrexate (MTX)	Apoptosis	T. Elango, et al., 2017 [45]
Vitamin D	Apoptosis	P. De Haes, et al., 2004 [51]
Calcipotriol	Apoptosis	R. Tiberio, et al., 2009 [52]
Efalizumab and Etaracizumab	Anoikis	J. Mei, et al., 2024 [59]
R-7-Cl-O-Necrostatin-1 (Nec-1s) and Necrosulfonamide (NSA)	Necroptosis (Animal model)	X. Duan, et al., 2020 [66]
Dimethyl fumarate (DMF)	Necroptosis	M. Burlando, et al., 2023 [67] M. Corazza, et al., 2021 [68] F.-l. Shi, et al., 2023 [70]
Saracatinib	Necroptosis	J. Li, et al., 2024 [71]
Disulfiram	Pyroptosis	X.-m. Hu, et al., 2024 [87] J. J. Hu, et al., 2020 [88]
Nicotinamide and nicotinamide mononucleotide	Parthanatos	Z. Zhang, et al., 2024 [98] M. El-Khalawany, et al., 2022 [99]
Ferrostatin-1 (Fer-1)	Ferroptosis (Animal model)	Y. Shou, et al., 2021 [105]
Glutathione	Ferroptosis	Nisha Kundu, et al., 2022 [106] R. Prussick, et al., 2013 [107]
Selenium	Ferroptosis	M. Nazıroğlu, et al., 2012 [108] I. G. Chambers and R. R. Ratan, 2024 [109]
Coenzyme Q10	Ferroptosis	G. A. Al-Oudah, et al., 2022 [110] Z. Kharaeva, et al., 2009 [111] K. Hadian, 2020 [112]
Deferoxamine	Ferroptosis	E. Abboud, et al., 2024 [114]
Myeloperoxidase inhibitor	NETosis	S. D. Neu, et al., 2021 [128]
Protein arginine deiminase 4 (PAD4) inhibitors	NETosis (Animal model)	C. Gajendran, et al., 2023 [129] J. Czerwińska, et al., 2022 [130]

## Data Availability

Data are contained within the article.

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
