# Peer review of "How Cells Die in Psoriasis?"

_ijms, 2025, doi:10.3390/ijms26083747_

Round 1
Reviewer 1 Report
Comments and Suggestions for Authors
The ami of the paper was to investigate the diverse modes of cellular death in psoriatic lesions, particularly focusing on keratinocytes, and their implications for disease progression and possible treatment strategies. The review has been thoroughly desinged, condeucted and the dta comprehensively described. This extensive analysis elucidates the complex interplay of multiple programmed cell death mechanisms in psoriasis pathogenesis. The identification of these distinct death pathways and their molecular signatures provides crucial insights into the multifaceted nature of psoriatic inflammation. The demonstrated involvement of various mechanisms underscores the complexity of cellular death processes in psoriatic lesions. Furthermore, the authors point out that understanding these intricate mechanisms may lead to more effective combination therapies for enhanced clinical outcomes in psoriasis management. The cited papers are a huge collection, however one is missing and it'd be advisable to add when describing gasdermin D in psoriatic lesions - https://pubmed.ncbi.nlm.nih.gov/37685853/
Author Response
Comment 1: The cited papers are a huge collection, however one is missing and it'd be advisable to add when describing gasdermin D in psoriatic lesions - https://pubmed.ncbi.nlm.nih.gov/37685853/
Response 1: We sincerely appreciate the thoughtful and constructive feedback provided on our manuscript investigating diverse modes of cellular death in psoriatic lesions. Regarding your specific recommendation about the gasdermin D discussion, we fully agree with your suggestion. We have amended our manuscript to include the reference you highlighted on line 258, which indeed provides valuable insights into gasdermin D's role in psoriatic skin lesions. This addition strengthens our analysis and ensures our literature review is more comprehensive. Thank you once again for your careful review and for helping us improve the quality and completeness of our manuscript.
Reviewer 2 Report
Comments and Suggestions for Authors
It is known that psoriasis, a chronic inflammatory dermatosis mediated by T lymphocytes, is the result of heterogeneous etiopathogenic processes. The intertwining of numerous cellular and molecular mechanisms acting on genetic predisposition determines the Th1/Th2 imbalance. This balance controls cell death and chronic inflammatory infiltrate.
- Keratinocytes are the main targets of psoriatic pathogenesis. In this context, the authors emphasize the role of keratinocyte cell death control mechanisms in the therapeutic management of psoriasis. To avoid any confusion I recommend the authors to clarify the information on lines 38-40 - ”autophagy, apoptosis, anoikis, ne-croptosis, pyroptosis, PANoptosis, parthanatos, ferroptosis, cuproptosis and NETosis ” - are forms/patterns of programmed cell death.
- Both extrinsic and intrinsic programmed cell death pathways are activated and controlled by a number of proinflammatory cytokines. Among these is the IL-17/Il-23 axis - an axis that explains the ”molecular fingerprint” of psoriatic plaques. Therefore, I recommend supplementing the information with the impact of IL-23 on cell death via resident memory T lymphocytes (TRM). Treg/TRM balance is necessary.
Author Response
Comment 1: Keratinocytes are the main targets of psoriatic pathogenesis. In this context, the authors emphasize the role of keratinocyte cell death control mechanisms in the therapeutic management of psoriasis. To avoid any confusion I recommend the authors to clarify the information on lines 38-40 - ”autophagy, apoptosis, anoikis, ne-croptosis, pyroptosis, PANoptosis, parthanatos, ferroptosis, cuproptosis and NETosis ” - are forms/patterns of programmed cell death.
Response 1: We sincerely thank for this valuable recommendation regarding the clarification of cell death terminology. We would like to respectfully note that our manuscript already explicitly identifies these pathways as "programmed cell or regulated death" on line 39. However, we fully appreciate the reviewer's concern about potential confusion, and have therefore further emphasized this classification on line 41, reiterating that all the mechanisms enumerated (autophagy, apoptosis, anoikis, necroptosis, pyroptosis, PANoptosis, parthanatos, ferroptosis, cuproptosis, and NETosis) represent forms of programmed cell death. This addition strengthens our analysis and ensures our literature review is more comprehensive.
Comment 2: Both extrinsic and intrinsic programmed cell death pathways are activated and controlled by a number of proinflammatory cytokines. Among these is the IL-17/Il-23 axis - an axis that explains the ”molecular fingerprint” of psoriatic plaques. Therefore, I recommend supplementing the information with the impact of IL-23 on cell death via resident memory T lymphocytes (TRM). Treg/TRM balance is necessary.
Response 2: We sincerely appreciate the reviewer's insightful recommendation regarding the IL-17/IL-23 axis and its relationship to programmed cell death pathways. We concur with the reviewer's assessment of these cytokines' critical importance in the molecular pathogenesis of psoriasis.
The survival of resident memory T lymphocytes (TRM) is dependent of IL-23 (Whitley, Sarah K et al. “Local IL-23 is required for proliferation and retention of skin-resident memory TH17 cells.” Science immunology vol. 7,77 (2022): eabq3254. doi:10.1126/sciimmunol.abq3254). Future research priorities should encompass the studies of the roles of the proinflammatory cytokines on cell survival in psoriasis, development of combination therapies targeting multiple death pathways, investigation of the temporal sequence of death mechanisms during disease progression, identification of pathway-specific biomarkers for personalized treatment approaches, and exploration of novel therapeutic agents targeting emerging pathways such as cuproptosis, while acknowledging current methodological limitations including the challenge of distinguishing between primary and secondary effects of pathway activation and the potential oversight of significant synergistic effects due to the predominant focus on individual pathways rather than their interactions.
Thank you once again for your careful review and for helping us improve the quality and completeness of our manuscript.